# Accelerating the prediction of $CO_2$ capture at low partial pressures in metal-organic frameworks using new machine learning descriptors

Ibrahim B. Orhan[1,2], Tu C. Le [3,4✉], Ravichandar Babarao [1,2,4✉] & Aaron W. Thornton [2,4✉]

Metal-Organic frameworks (MOFs) have been considered for various gas storage and separation applications. Theoretically, there are an infinite number of MOFs that can be created; however, a finite amount of resources are available to evaluate each one. Computational methods can be adapted to expedite the process of evaluation. In the context of $CO_2$ capture, this paper investigates the method of screening MOFs using machine learning trained on molecular simulation data. New descriptors are introduced to aid this process. Using all descriptors, it is shown that machine learning can predict the $CO_2$ adsorption, with an $R^2$ of above 0.9. The introduced Effective Point Charge (EPoCh) descriptors, which assign values to frameworks' partial charges based on the expected $CO_2$ uptake of an equivalent point charge in isolation, are shown to be the second most important group of descriptors, behind the Henry coefficient. Furthermore, the EPoCh descriptors are hundreds of thousands of times faster to obtain compared with the Henry coefficient, and they achieve similar results when identifying top candidates for $CO_2$ capture using pseudo-classification predictions.

[1] School of Science, Centre for Advanced Materials and Industrial Chemistry (CAMIC), RMIT University, Melbourne, VIC 3001, Australia. [2] CSIRO Future Industries —Manufacturing Business Unit, Clayton, VIC 3169, Australia. [3] School of Engineering, RMIT University, Melbourne, VIC 3001, Australia. [4] These authors jointly supervised this work: Tu C. Le, Ravichandar Babarao, Aaron W. Thornton. ✉email: tu.le@rmit.edu.au; ravichandar.babarao@rmit.edu.au; aaron.thornton@csiro.au

Since the industrial revolution, atmospheric $CO_2$ levels have risen more than 140 ppm, recording measurements above 420 ppm as of June 2022[1]. This increase in $CO_2$ concentration in the atmosphere has raised questions regarding the ramifications of such a drastic change; it was found that ~60% of the global warming effects being attributable to $CO_2$ emissions[2]. Not only has the increase in $CO_2$ concentrations been proven to have impacts on the climate, but it also has potentially negative effects on mammalian physiology[3]. Carbon capture and storage (CCS) technologies will play a role in offsetting the accumulation of this gas and thus negate the drawbacks of using carbon-intensive technologies.

Similar to other pollutants, the key advances in $CO_2$ capture technology will likely stem from the adoption of CCS as a standard practice for all large stationary fossil fuel installations[4]; however, the cost of CCS currently remains a major consideration. Finding commercially viable end-use opportunities for the captured $CO_2$ is still a growing interest as it is expected that CCS will mitigate 14–20% of total anthropogenic $CO_2$ emissions by 2050[5]. Therefore, for CCS to be economically viable, either the cost of implementing the technology must be minimized, or the captured $CO_2$ must be commercially useful.

In the context of $CO_2$ capture, alongside zeolites, activated carbon, and others, metal-organic frameworks (MOFs), which are structures composed of metal oxide clusters connected through organic linkers, are gaining traction as candidate materials[6]; it has been demonstrated that MOFs can be adapted to pellet or film forms without losing their sorption properties[7] and can therefore be more readily adapted to $CO_2$ capture from flue gases. The malleability of MOFs while retaining their sorption properties is an advantage for configuring them into forms that can be better suited for large-scale CCS[8]. Beyond their malleability, MOFs have been extensively studied in pre- and post-combustion $CO_2$ capture applications[9] as well as being studied for direct air capture (DAC)[10].

Unlike the CCS options that require access to the source of $CO_2$, DAC does not require direct access to the $CO_2$ source. However, this method faces its own set of challenges. The reduced concentration of $CO_2$ compared to the concentrations at point sources of $CO_2$, as well as $H_2O$ having a greater partial pressure at atmospheric conditions result in many adsorbents preferentially adsorbing $H_2O$ over $CO_2$[11]. To improve the performance of these materials, new configurations of MOFs such as multivariate MOFs are being developed for enhancing the separation of $CO_2$ from various gases[12]. With endless new variations to the MOF family of materials being added, it would be nearly impossible to evaluate the entire MOF-space for their $CO_2$ capacity. As a method of faster evaluation, machine learning (ML) models trained on high-throughput molecular simulation data can be used.

As ML becomes an increasingly popular tool in various scientific fields[13–17], its applications with respect to predicting gas adsorption and separation properties in MOFs continue to expand[18–20]. Using a classification model, Aghaji et al. rapidly identified MOFs for methane purification with high $CO_2$ uptake and high selectivity[21]. The ML model was built using geometrical descriptors and the aim was to determine MOFs with $CO_2/CH_4$ selectivity higher than 5 or higher than 10 and to determine MOFs with 2 or 4 mmol g$^{-1}$ working capacity or greater. Evaluating their model through a receiver-operator curve, the area under the curve was shown to reach 0.95 with the missed true-positives appearing at the lower-performance end of the spectrum.

Using the Topologically Based Crystal Constructor[22], Anderson et al., computationally constructed 400 MOF crystals. Density functional theory (DFT) calculations were performed to optimize the adsorbate binding configurations; then grand canonical Monte Carlo (GCMC) simulations were run using the RASPA package[23]. Unlike the low partial pressure in DAC conditions, the authors simulated the adsorption of $CO_2$ both as mixtures (with $H_2$ and $N_2$) and in pure form. Using six ML learning models they were able to obtain the coefficient of determination, $R^2$, as high as 0.905 and gain insight into the importance of descriptors used in the models. Expanding on the computational methods utilized, they also demonstrated that genetic algorithms could search for characteristics that correlate to the highest predicted uptakes and selectivities in the ML model. Their work demonstrated that machine learning based on simple descriptors can be an effective simulation-free tool to predict $CO_2$ capture metrics while highlighting the need for different design strategies to optimize various MOF metrics.

As ML applications related to $CO_2$ capture in MOFs are proving to be plausible, the question of whether they will be effective in predicting $CO_2$ capture in DAC conditions remains. In this paper, descriptors are developed to better model MOFs and train ML models. Using various combinations of these features, the ML algorithms were used to identify the most influential descriptors in yielding accurate predictions to find the best candidate for $CO_2$ capture. Consideration of hydrophobicity must be given because the partial pressure of $H_2O$ in air is typically much greater than that of $CO_2$ in DAC conditions. In this study, three concentrations of $CO_2$ were considered: 40 Pa, 1 and 4 kPa. These values correlate with the concentrations in air and indoor settings, manned spacecraft, submarines, and emergency rebreathers for diving and mining applications[24–29]. Danaci et al. highlight the limitation of looking only at the capacity and selectivity of MOFs. They emphasize the necessity of also investigating the rate of mass transfer and the ease at which the adsorbent can be regenerated under moderate conditions[30]. While the rate of mass transfer and regeneration conditions of adsorbents were not studied in this paper, future research may build on the findings of this paper and utilize the new descriptors to predict these aspects of MOFs.

## Methodology

**Dataset curation.** In this study, MOFs from the CoRE MOF dataset (3378 structures) and the Anion-pillared MOF dataset (936 structures) were used, where partial charges on the atom sites had been calculated based on DFT using the DDEC method[31,32]. While both datasets were used in the ML model, only the Anion-pillared MOFs were used to estimate the necessary time for gathering descriptors. To allow the ML model to be fitted to a wider range of MOF structures, screening based on features was not conducted. Typically, descriptors of a dataset in ML are multi-dimensional and can be separated into distinct groups based on the nature of their measurement. In this dataset, the descriptors (Table 1) are categorized into atom type (A), geometric (B), chemical (C), effective point charge (D), and energy (E). Each category of descriptor carries multiple dimensions relating to measurements taken on MOFs, such as the number of specific atoms found in a unit cell or the size of pores. The atom type, geometric, and chemical descriptors have already been shown to be effective in building ML for predicting other gas-related properties of MOFs[20]. The dataset can be found in Supplementary Data 1.

**Monte Carlo simulations.** The machine learning was built to predict a target variable ($CO_2$ uptake) that was simulated using the grand canonical Monte-Carlo (GCMC) method. Separate ML models were built for each partial pressure of interest. For each ML model, the target variable ($CO_2$ uptake) was gathered through GCMC simulations using the RASPA package[33]. The GCMC simulations were run such with the cutoff distance for interactions set to 12.5 Å, for 20,000 cycles, at a temperature of 298 K. The universal forcefield (UFF)[34] was used for the van der Waals (VDW) parameter of the framework atoms. The $CO_2$ molecule[35]

**Table 1 Descriptor groups in the dataset.**

| Group | Descriptor |
|---|---|
| Dataframe skeleton | MOF name |
| | Target variable $CO_2$ uptake (mmol g$^{-1}$) |
| | Pressure |
| Atom type (A) | Number of H atoms per unit volume |
| | Number of C atoms per unit volume |
| | Number of N atoms per unit volume |
| | Number of F atoms per unit volume |
| | Number of Cl atoms per unit volume |
| | Number of Br atoms per unit volume |
| | Number of V atoms per unit volume |
| | Number of Cu atoms per unit volume |
| | Number of Zn atoms per unit volume |
| | Number of Zr atoms per unit volume |
| Geometric (B) | Accessible surface area |
| | Non-accessible surface area |
| | Accessible volume |
| | Non-accessible volume |
| | Accessible probe-occupiable volume |
| | Non-accessible probe-occupiable volume |
| | Pore limiting diameter |
| | Largest cavity diameter |
| | Largest free path diameter |
| | Density |
| | Volume |
| Chemical (C) | Total degree of unsaturation |
| | Metallic percentage |
| | Oxygen to metal ratio |
| | Electronegative to total ratio |
| | Weighted electronegativity per atom |
| | Nitrogen to oxygen ratio |
| Effective point charge (D) | Charge-based uptake at 40 Pa |
| | Charge-based uptake at 1 kPa |
| | Charge-based uptake at 4 kPa |
| | Charge-based uptake at 40 Pa averaged per atom |
| | Charge-based uptake at 1 kPa averaged per atom |
| | Charge-based uptake at 4 kPa averaged per atom |
| | Charge-based uptake at 40 Pa per unit volume |
| | Charge-based uptake at 1 kPa per unit volume |
| | Charge-based uptake at 4 kPa per unit volume |
| Energy descriptor (E) | Henry coefficient |

Descriptors groups in the dataset where features are grouped based on similarities into atom type (A), geometric (B), chemical (C), effective point charge (D), and the energy descriptor (E). The dataset is curated by MOF name and the pressure at which the simulation was performed. The corresponding $CO_2$ adsorption is recorded for the simulation result of each MOF and pressure combination.

was assigned translation, rotation, reinsertion, and swap probabilities of 0.5, 0.5, 0.5, and 1, respectively. Partial charges calculated previously using DFT based on DDEC methods were used to compute the Coulomb interactions[32]. Any simulations that exceeded 24 h or that yielded errors were discarded. The same $CO_2$ molecule parameters were used when running simulations to gather the Henry coefficients. The parameters used to define the $H_2O$ molecule[36] and $CO_2$ molecule are presented in Supplementary Notes 1, 2.

**Machine learning models**. The random forest (RF) algorithm, which dates back to 1995, has been proven successful in various contexts and has seen various changes since its first proposal[37].

The algorithm, which utilizes numerous decision trees, is able to divide the search space at the nodes of trees. A drop in information entropy at each traversed node of the decision tree can provide insight into which descriptors are most important in their respective context. The bagging method allows the algorithm to further capture nuances by using different samples of the data to build the trees that make up the forest. The robustness that has been proven successful and the ability to gather insight into the features were the rationale for the decision to use RF as the algorithm for the model. The SciKit-Learn module[38] was used to access this algorithm. The coefficient of determination *r2_score* ($R^2$) and the root of the *mean_square_error* function (RMSE), present in the same module, were used to evaluate the mode

**Geometric and energy descriptors**. The Henry coefficient ($K_H$) represents how strongly a gas molecule interacts with an adsorbent. As such, an ML model could significantly benefit from including the Henry coefficient as a descriptor. The calculations are similar to the GCMC simulations described above. However, instead of specifying a specific pressure, the simulation is run by setting the Widom probability to 1 and by including the ideal gas Rosenbluth weight in the RASPA simulation parameters.

The Henry coefficient was calculated for two gases: $H_2O$ and $CO_2$. The results from the $CO_2$ simulations were directly included in the ML model, while the $H_2O$ results were used to evaluate the hydrophobicity of the candidate materials. The $H_2O$ model was derived from TIP4P.

The geometric descriptors are gathered using Zeo++[39], while the energy descriptors are gathered through additional molecular simulation. The probe radius in Zeo++ was set to be approximately the size of $CO_2$ at 1.5 Å. Pore volumes were calculated using 50,000 sample points, while surface areas were calculated using 2000 sample points. The energy descriptor was calculated using a molecular simulation package RASPA[33]. 20,000 cycles were simulated at a temperature of 298 K with a Widom probability of 1 to determine the Henry coefficients. The remaining categories of descriptors were gathered through in-house developed scripts (outlined in the Supplementary Discussion).

**Effective Point Charge (EPoCh) descriptors**. The EPoCh descriptors aim to quantify the effects of atomic partial charges found within the MOF structure. In order to quantify the influence of partial charges in the absence of VDW interactions, the descriptor represents the equivalent uptake with respect to equivalent point charges.

By constructing a single hypothetical atom in RASPA, which contains no VDW interactions and no mass, $CO_2$ uptake was simulated at various pressures and charges. See Fig. 1 depicting the snapshots from simulations with varying pressure and charges. The atom was assigned values ranging between −5e to +5e and the pressures simulated were primarily below 0.1 bar. For a negatively charged atom, the positively charged carbon atoms in the $CO_2$ molecule are attracted to the site. For a positively charged atom, the negatively charged oxygen atoms in the $CO_2$ molecule are attracted to the site. The stronger the charge, the higher the $CO_2$ uptake. $CO_2$ uptake also increases with pressure, though at different rates for the different charges.

By gathering sampled simulation data of the search space, a 2D surface indicating the resulting adsorption in a 3D space could be plotted where the remaining two axes are the corresponding pressure and charge of the atom in the simulation (demonstrated in Supplementary Fig. 1). The surface was fitted to the simulation

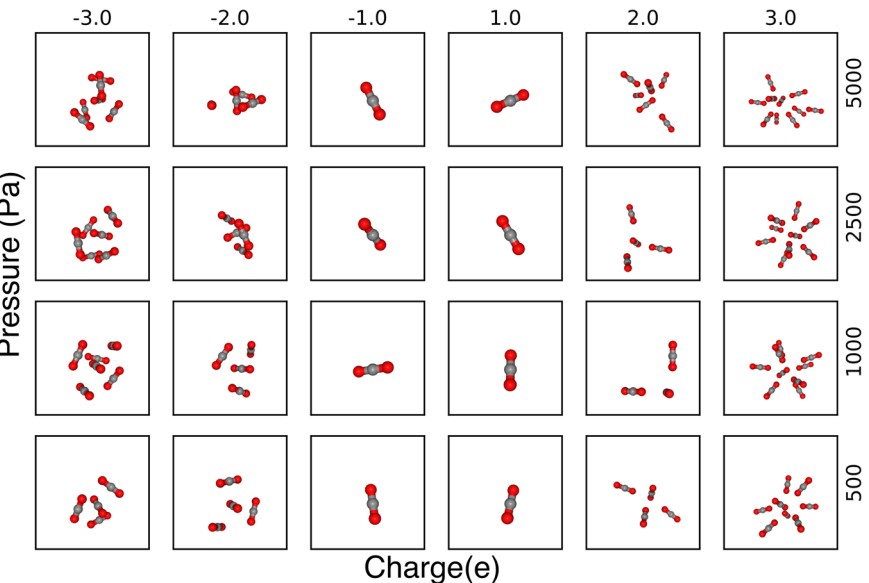

**Fig. 1 Pressure–charge effects in the Effective Point Charge (EPoCh) simulations.** Snapshots from the molecular simulations of $CO_2$ uptake around a single hypothetical atom with varying pressures and charges.

results using the following equation:

$$f(Q, p) = \alpha_1 Q + \alpha_2 Q^2 + \alpha_3 Q^3 + \alpha_4 Q^4 + \alpha_5 Q^5 + \alpha_6 Q^6 + \alpha_7 Q^7 \\ + \alpha_8 p + \alpha_9 p^2 + \alpha_{10} p^3 + \alpha_{11}$$
(1)

where $Q$ is the partial charge, $p$ is the partial pressure and $\alpha$ are the fitted coefficients (listed in SI1.1–SI1.2). To calculate the descriptor of a framework, $f$ is calculated for every atom, $i$, within the framework and averaged, as follows:

$$E_i = \max(0, f_i(Q_i, p))$$
$$E_{ave} = \sum_{i=1}^{N} \frac{E_i}{N},$$
(2)

where $E_i$ is the estimated uptake (mol cm$^{-3}$) for a charged atom $i$, and $E_{ave}$ is the averaged uptake over $N$ atoms within a framework. To prevent values below zero from being included, any evaluation of parameters that yield a subzero output from $f$ is set to zero. The partial charges on the atoms are unique to each framework and by evaluating each charge of a framework through Eq. (2), their isolated effects are estimated. The results are averaged both volumetrically and atom-wise, to determine a suite of EPoCh descriptors for each MOF structure.

## Results and discussion

The complete dataset yielded 12,637 simulation results for the three pressures simulated; 4243 of the simulation results were at 0.4 mbar, 4186 at 0.1 mbar, and 4208 at 0.04 mbar. The differences between the number of datapoints for the different pressure settings are a result of the simulations which were terminated after 24 h.

Figure 2 depicts the correlations between the descriptors and the target variable, $CO_2$ uptake. An inspection of the gathered data indicates that a number of the EPoCh descriptors have the highest correlation with uptake. This is closely followed by a number of chemical descriptors, after which the descriptors are either no longer correlated with the target variable or the correlation ($r$) becomes negative. No descriptor displayed a significant negative correlation with the target variable. Intuitively, there are highly positive and highly negative correlations between descriptors of the various categories. For example, the various correlations between

pore diameters and the averaged EPoCh descriptors are observed. Interestingly, the EPoCh descriptors show highly positive or highly negative correlations with a number of atom types and chemical descriptors. This is likely a result of the partial charges that arise from certain atoms in the framework.

**Henry's law**. Henry's law is used to estimate the uptake of gases at low partial pressures where uptake, $U$, is calculated as the product of Henry coefficient and pressure, $K_H p$. If the computationally intensive Henry coefficient has already been obtained, it is possible that Henry's Law would negate any need for machine learning. The distribution shown in Fig. 3a of Henry's coefficient values indicates that there is a wide range of values in the dataset. While some MOFs with exceptionally high Henry's coefficient values could clearly be considered outliers, there are, however, some MOFs that show high uptake and should not be ignored. The histogram shows the distribution of Henry's coefficient values after taking the natural log, and the distribution of values closely resembles a bell curve with a slight right skewness. The skewness following the log-transformation is 1.21 compared to 97.54 prior. The range of Henry's coefficient values can be highlighted by the maximum value being $10^{30}$ times greater than the minimum value, leading to a dataset where the mean Henry coefficient is $1.46 \times 10^3$ while the median Henry coefficient is $2.64 \times 10^{-4}$, indicating a significant range of values present in the dataset. The presence of high Henry coefficient values would signal that those MOFs demonstrate a sharp increase in adsorbed $CO_2$ at low pressures. Figure 3b shows the simulated isotherms of the MOFs with the highest uptake along with the Henry Law prediction. There is some agreement at ultra-low pressures of around 10 Pa, however, the simulated isotherms quickly move outside the linear region of Henry's Law. Considering that this study is focused on uptakes above 40 Pa, the Henry Law may not be applicable.

Figure 4 presents the performance of the Henry Law model for MOFs with Henry's coefficients below or equal to 0.001 (Fig. 4a), and MOFs with Henry's coefficients between 0.001 and 1 (Fig. 4b). Using the full dataset there is an unacceptable $R^2$ of $-9.807 \times 10^{15}$. At high pressures and high Henry's coefficient values, there is no observable trend (Fig. 4b). By looking at lower Henry coefficient values, the trend becomes more observable (Fig. 4a). Limiting the dataset to MOFs below 0.001 Henry's coefficient, at 40 Pa there is

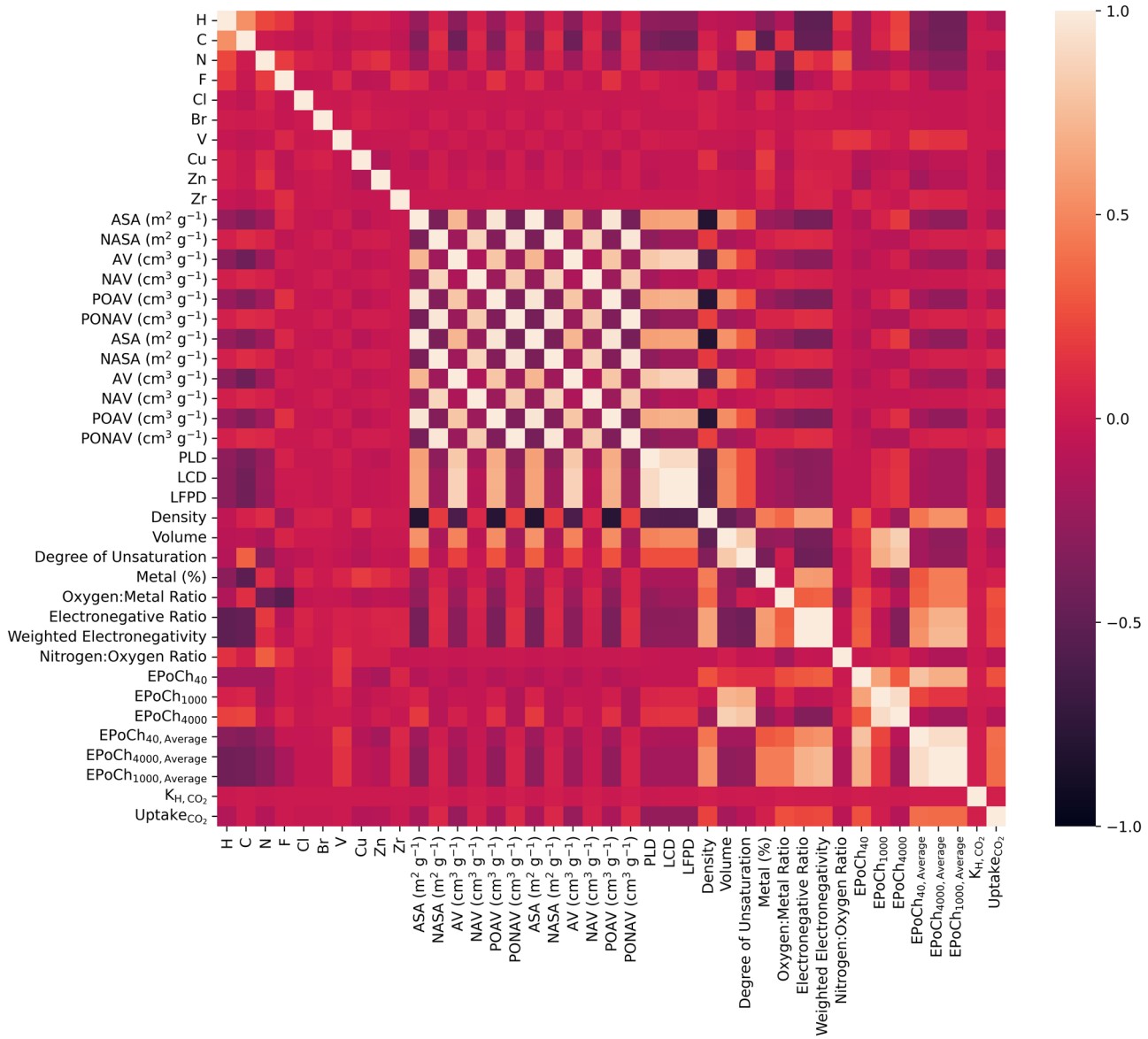

**Fig. 2 Correlation within the dataset visualized in a heatmap.** Pearson correlation ($r$) heat-map of descriptors and target variable where the correlation between each pair of features can be found at the intersection of their respective column–row intersection. The values of the correlation are color-coded according to the scale shown to the right of the heat-map.

good agreement between the GCMC uptake and the Henry's Law uptake. For this case, the correlation obtains a $R^2$ of 0.98. Similarly, for 1000 Pa, there is a good correlation with an $R^2$ of 0.924; however, there is negligible uptake at 40 Pa and low uptake (<1 mmol g$^{-1}$) at 1000 Pa. At 4000 Pa, MOFs display reasonably high uptakes (up to 4 mmol g$^{-1}$), however, the agreement between GCMC results and Henry's Law diminishes with an $R^2$ of 0.206. As Henry's coefficient value increases, the adherence to Henry's Law diminishes due to the MOFs reaching their saturation points at lower pressures. Therefore, Henry's Law is a poor physical model for identifying candidates with high uptakes (>1 mmol g$^{-1}$) at low partial pressures (<4000 Pa).

While the Henry coefficient is an influential descriptor for building accurate machine learning models, the direct calculation of uptake does not appear to be possible through Henry's Law alone. The correlation between uptake calculated through Henry's Law and uptake simulated in GCMC diminishes as the pressure and/or Henry's coefficient value increases. This means that it is possible to predict, with high accuracy, the MOFs that have lower

uptake, while the MOFs that have significantly greater Henry coefficient values do not fit the law at the pressures considered. This suggests that those MOFs have such a high proclivity to capturing $CO_2$ that they have already surpassed the linear region of the isotherm where Henry's Law holds true. These are the exact MOFs that are of most interest for $CO_2$ capture and thus resorting to other methods, such as ML, are worthwhile pursuit.

**Machine learning models.** By splitting the dataset into 80% training and 20% testing, the model is evaluated (Fig. 5). The combination of feature groups A, B, and C acted as a benchmark model with an $R^2$ of 0.541 for 40 Pa. The influence of the additional descriptors on the model's performance was evaluated. At each pressure, the model with EPoCh descriptors (D) combined with the benchmark descriptors outperformed the benchmark model, e.g., $R^2$ of 0.715 for 40 Pa. An increase in performance, $R^2$ of 0.916 for 40 Pa, is observed when including the Henry coefficient energy descriptors (E). The ML model incorporating all descriptors in

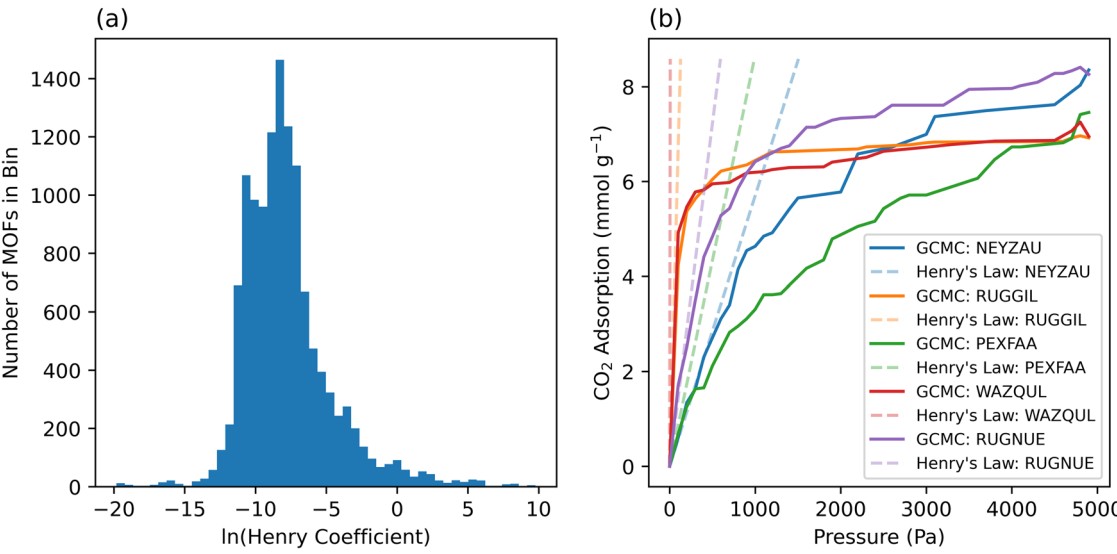

**Fig. 3 Overview of Henry coefficient values in the dataset and isotherm–Henry's law comparison for MOFs with greatest CO$_2$ adsorption.**
**a** Distribution of Henry coefficient values for the MOF dataset. **b** Uptake versus pressure for the MOFs with the highest adsorption, highlighting the non-linearity that Henry's Law does not model.

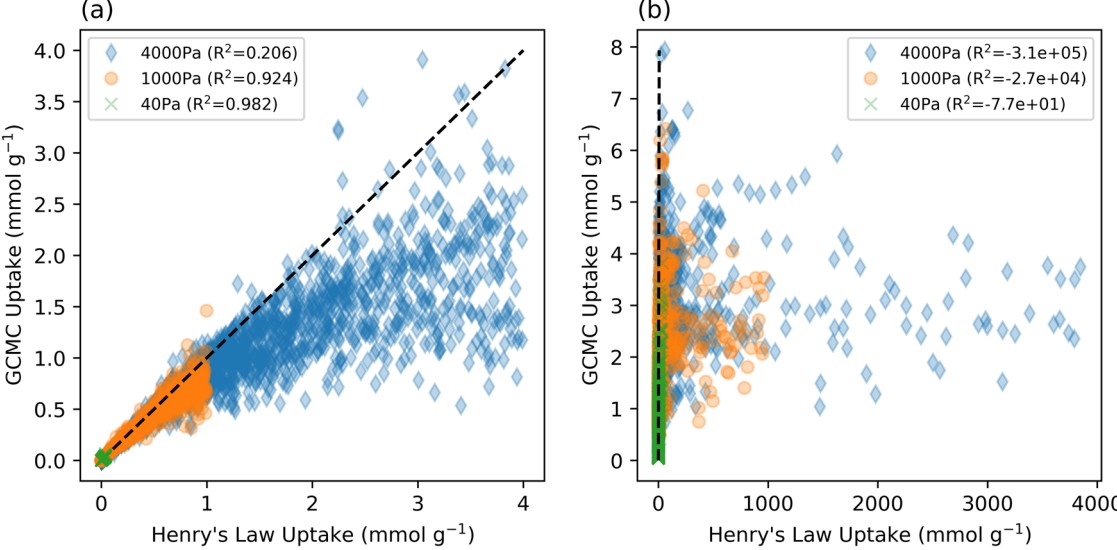

**Fig. 4 Henry's Law applied to the dataset.** GCMC uptake versus Henry's Law uptake for **a** MOFs with Henry's coefficients below or equal to 0.001, and **b** MOFs with Henry's coefficients between 0.001 and 1. The dashed black lines indicate perfect agreement between GCMC and Henry's Law.

unison, was able to yield predictions where the $R^2$ surpasses 0.9 for all pressures. Root mean squared errors (RMSE) reveal the same trends and can be found in Supplementary Table 1.

As the computational expense of obtaining the Henry coefficients of MOFs is non-negligible, there are benefits to analyzing the ML model without these descriptors. When energy descriptors are not included, there is a decrease in the performance of the model. However, the trends between predictions and their corresponding simulation values are still observable with $R^2$ values ranging between 0.69 and 0.742 (see Figs. 6 and 7).

For a MOF to be considered successful in DAC, an uptake criterion of at least 1 mmol g$^{-1}$ was set. Using this as a criterion for classification, the ML model can ultimately act as a screening method to determine which candidates require in-depth analysis. Here a pseudo-classification method is introduced, where MOFs that are predicted to have an uptake above a certain threshold would either be simulated in detail or synthesized experimentally, while the rest are discarded. Setting this threshold value to 1 mmol g$^{-1}$, a

*positive* label would be given to MOFs that are predicted to have CO$_2$ adsorption equal to or greater than this value. Through such a method, the regression model ML algorithm can be turned into a pseudo-classification method. The predicted versus simulated plots can then be separated into quadrants divided by vertical and horizontal lines at the threshold values (see Figs. 6 and 7). These quadrants would correspond to the true negative predictions (bottom-left), true positives (top-right), false negatives (bottom-right), and false positives (top-left).

From this information, the metric can be calculated to assess the performance of the ML models. For example, the recall metric can be calculated, which is the number of true positives divided by the addition of false negatives and true positives. This is a measure of sensitivity where a higher recall means that the model is capturing the relevant information. In other words, a higher recall means that the model is good at determining the number of positive candidates. Additionally, the precision metric can be calculated, which is the number of true positives divided by the

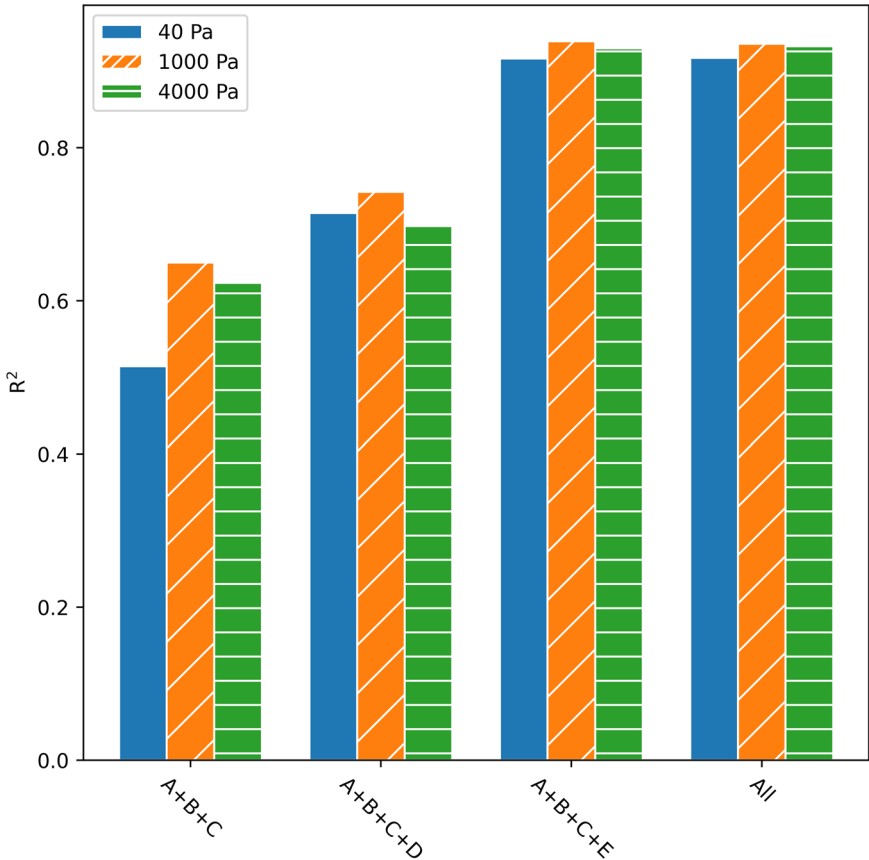

**Fig. 5 Performance metrics of feature group combinations.** Coefficient of determination $R^2$ for the ML models. A + B + C is the benchmark model using conventional descriptors. The addition of the EPoCh descriptors (D) and the Henry coefficient energy descriptors (E) shows an improvement in the model.

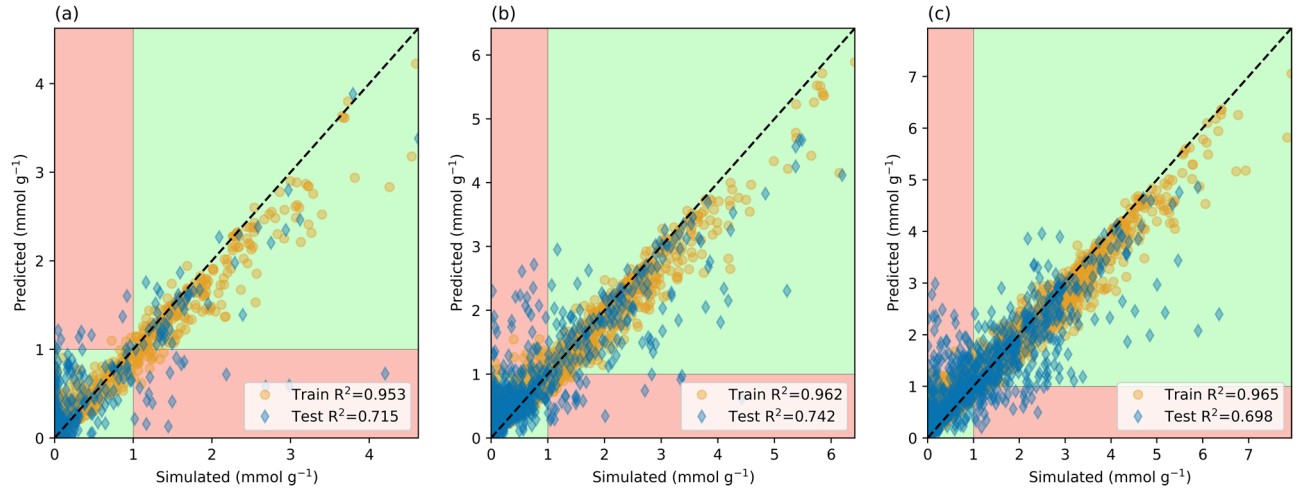

**Fig. 6 Performance of the ML model using descriptors excluding the Henry Coefficient.** Predictions from a model built without energy descriptors of $CO_2$ uptake at **a** 40 Pa, **b** 1 kPa, **c** 4 kPa where the x-axis corresponds with the simulated values and the y-axis is the predicted adsorption; the dashed line indicates a perfect coefficient of determination ($R^2 = 1$). The green regions demonstrate correct pseudo-classification results, while the red regions demonstrate predictions that would be misclassified.

addition of false positives and true positives. Precision indicates a level of accuracy such that the model should minimize the number of false positives. A high precision means that resources are not unnecessarily wasted on candidates that have low uptakes.

Using all descriptors, the model displayed recalls of 0.969, 0.975, and 0.983 at 40 Pa, 1, and 4 kPa, respectively. The corresponding precisions were 0.849, 0.914, and 0.952, respectively, for 40 Pa, 1, and 4 kPa. The recall rates indicate that at all

pressures, more than 95% of candidates with a simulated uptake above the 1 mmol $g^{-1}$ threshold were correctly predicted by the ML model. While the precisions indicate that at all pressures, more than 84% of the ML predictions above the 1 mmol $g^{-1}$ threshold were true positives.

In comparison, the ML model that does not incorporate the Henry coefficient is assessed. At 40 Pa, 1, and 4 kPa the recalls were 0.719, 0.838, and 0.883, respectively. The precisions were

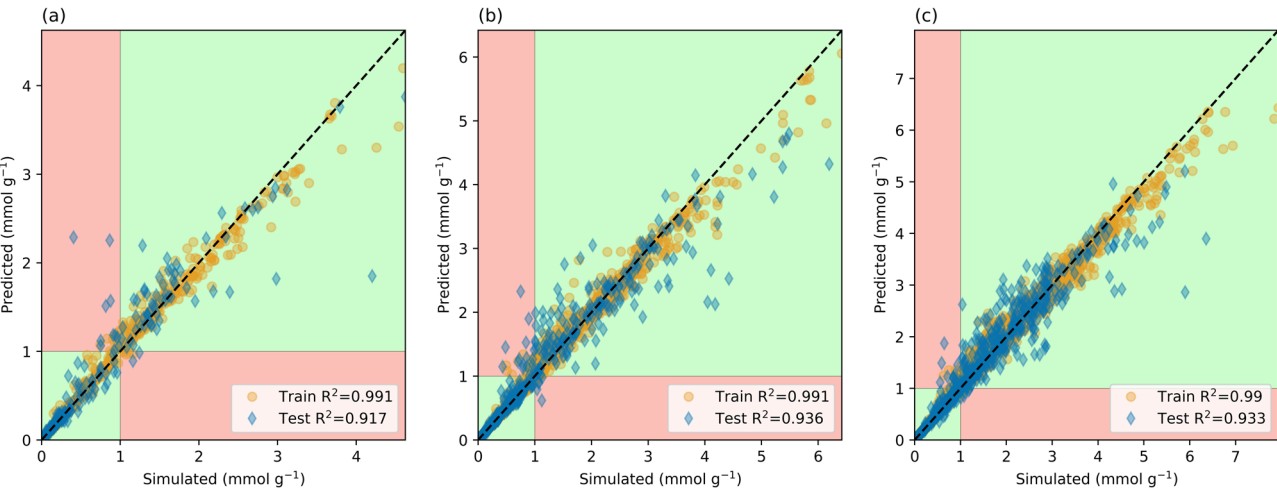

**Fig. 7 Performance of the ML model using all descriptors including the Henry Coefficient.** Predictions from a model built using all descriptors of $CO_2$ uptake at **a** 40 Pa, **b** 1 kPa, **c** 4 kPa where the *x*-axis corresponds with the simulated values and the *y*-axis is the predicted adsorption; the dashed line indicates a perfect coefficient of determination ($R^2 = 1$). The green regions demonstrate correct pseudo-classification results, while the red regions demonstrate predictions that would be misclassified.

0.807, 0.778, and 0.791, respectively, for 40 Pa, 1 and 4 kPa. Despite requiring significantly fewer computational resources to obtain the descriptors of this ML model, the recalls and precisions are similar to those of the ML model using the Henry coefficient descriptor. The precisions did not vary significantly between the ML models and there was only a 4 percentage points difference between the two models at 40 Pa. Therefore, despite the lower $R^2$ values of the EPoCh-based ML model, there is excellent performance in the classification of top candidates with uptakes above 1 mmol g$^{-1}$.

At 40 Pa (Fig. 6a), the largest root mean squared errors in the training set and test set predictions were 1.42 and 3.48 mmol g$^{-1}$, respectively; there were 64 MOFs with a $CO_2$ adsorption ≥1 mmol g$^{-1}$ in this set and the ML model predicted 73 to have an uptake greater than this value. At 1 kPa (Fig. 6b), 1.99 and 3.8 mmol g$^{-1}$ were the largest discrepancies between simulated and predicted values for the training set and test set respectively. At 1 kPa, 197 MOFs had a simulated $CO_2$ adsorption ≥1 mmol g;$^{-1}$ the ML model predicted there to be 209. Similarly, at 4 kPa (Fig. 6c), the largest differences between simulated and predicted values in the training set and test set were 2.03 and 3.96 mmol g$^{-1}$, respectively. There were 403 MOFs with simulated adsorption ≥1 mmol g;$^{-1}$ the ML model predicted there to be 418. In all pressures considered, the number of MOFs predicted to have $CO_2$ adsorption above the 1 mmol g$^{-1}$ threshold was greater than the true number above this threshold.

Since finding candidate materials to further analyze in a timely manner is the ultimate goal, the time necessary to gather each descriptor is also an important factor. Especially as the number of hypothetical MOFs continues to grow indefinitely. Figure 8 demonstrates the average necessary time (on a log scale) for gathering descriptors in each group. Starting with the brute-force approach where every candidate is subject to a complete GCMC simulation at the pressures of interest, the estimated time to assess 10,000 MOFs is $1.09 \times 10^8$ s (3.45 years). Clearly, this is not feasible when considering millions of candidates. The Henry coefficient is capable of improving the accuracy of the ML model, however, the time required to assess 10,000 MOFs is $3.27 \times 10^7$ s (~1 year). The approach is also not feasible. Fortunately, there is a significant reduction in computational time by orders of magnitude for the remaining descriptors. For example, the geometric descriptors would require 54,000 s (15 h) to assess 10,000 MOFs, followed by the EPoCh descriptors at 20 s and

finally the atom type descriptors at 3.7 s. Therefore, there are enormous benefits of accelerating the screening process by using the EPoCh descriptors in combination with atom type and geometric descriptors.

To further emphasize the advantage of the EPoCh descriptors, the $R^2$ and RMSE values are weighted according to the average time required to gather each descriptor. The model with EPoCh descriptors outperforms the models containing the Henry coefficient by a magnitude of over 450, shown in the adjusted $R^2$ in Supplementary Fig. 2. Displaying a similar performance for the time-weighted RMSE, the EPoCh descriptor again outperforms any other feature combination considered. This means if speed and accuracy are equally important, the ML model with EPoCh is ~30,000% better than the ML with Henry's coefficient.

In comparison with the Henry coefficient, the EPoCh descriptors provide additional information about the adsorption behavior specifically around charged atoms. The Henry coefficient gives an overall picture of the interactions for $CO_2$ uptake in the isotherm's linear region at infinite dilution. All interactions are captured in a single number $K_H$, and by multiplying it by the pressure, we can obtain the uptake. However, some high-uptake MOFs quickly (almost immediately) fall outside the linear region. The EPoCh descriptor, on the other hand, is indifferent to which region of the MOF's isotherm it falls, as it is calculated at different pressures. Unlike $K_H$, it does not capture all interactions; it captures only the electrostatic interactions at the pressures considered, ignoring the VDW interactions. Overall, the ML model is improved by the charged atoms' electrostatic interactions being modeled more precisely.

For both the evaluation of $CO_2$ uptake through GCMC and the calculation of $K_H$, the computational time necessary depends on the length of simulations run, the pressure considered (where applicable), and the size of the MOF; these simulations were conducted according to the specifications detailed in the "Methodology" section. While the energy grids are particularly beneficial when evaluating full isotherms, the decision regarding their use is another factor that influences the computational time. The calculation of energy grids, being computationally non-negligible, was not used in this study which considered only three pressure settings. The time requirements for the descriptors and target variable indicated in this section are based on the means of elapsed times while compiling the Anion-Pillared MOFs dataset on the Gadi High-Performance Computing Cluster of the

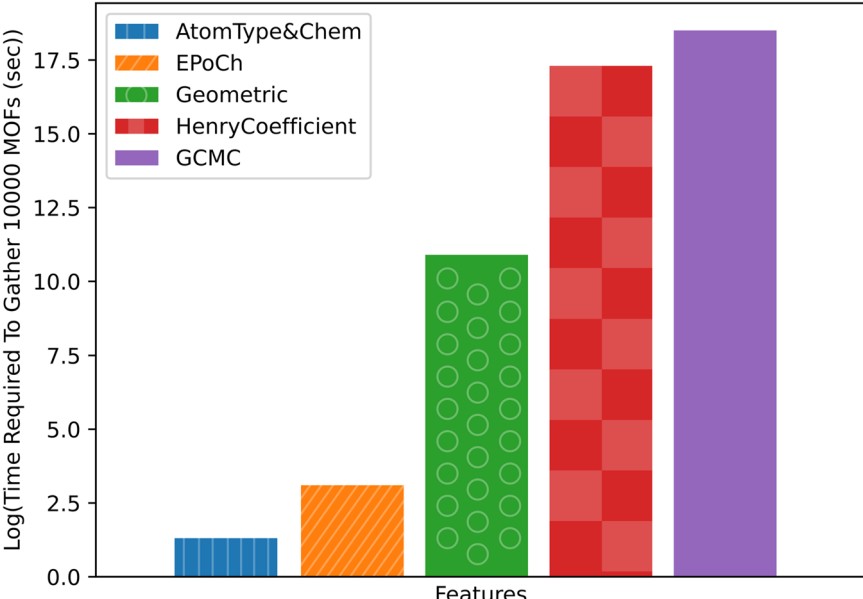

**Fig. 8 Time requirements of each descriptor group.** Estimated times necessary to gather descriptors of a 10,000 MOF dataset (based on timings gathered from the Anion-Pillared MOFs).

National Computational Infrastructure of Australia for GCMC, Widom Insertion, and Zeo++, and a PC with a 2.9 GHz 6-Core Intel Core i9 processor for the calculation of EPoCh, Chemical, and atom type descriptors.

**Feature importance**. When the Henry coefficient is used to build the ML model, the relative importance of the Henry coefficient in making predictions outweighs the relative importance of all remaining descriptors. Where the sum of relative importance for descriptors in groups A–D, on average, yields 0.17, the Henry coefficient yields a relative importance of 0.83 (where all features sum to 1). The influence this descriptor has can be further highlighted by comparing the performance of models where this descriptor is included to those where it is excluded. Keeping in mind that the models that incorporated this descriptor had $R^2$ values near 0.95, while those that did not incorporate this descriptor had $R^2$ values around 0.7.

It is evident, both through the performance of the models and through the ranking of feature importance, that the Henry coefficient displays the greatest influence on predicting $CO_2$ adsorption capacity. Though, it is not always possible or practical to run simulations to gather this group of descriptors, it is clear that the EPoCh descriptors would play a pivotal role in the absence of the Henry coefficient (see Fig. 9). Both the EPoCh and Henry coefficient descriptors have a drawback of requiring the charges of atomic sites. If these are readily available and are known to be accurate, the EPoCh descriptors could save considerable computational power.

So far, separate ML models have been created for each pressure considered. While this is beneficial for evaluating MOFs where there is already data available, the capacity of MOFs at pressures unseen by the model is not predictable through separate models. By adding pressure as a descriptor and combining all the datapoints to a single ML model, predictions were made on MOFs at pressures where there were previously no data. Two pressures were selected for testing predictions by interpolation (400 Pa) and extrapolation (10,000 Pa). 400 Pa was selected so that the model could make predictions between pressures on which it has been trained (40 and 1000 Pa). To see how it performs when considering pressures beyond the maximum

(4000 Pa), 10,000 Pa was selected. At 400 Pa the model yielded an $R^2$ of 0.54 while at 10,000 Pa the $R^2$ was 0.72 when the Henry coefficient was excluded from the dataset (shown in Supplementary Fig. 3). Applying the same pseudo-classification threshold of 1 mmol g$^{-1}$, the precision (ratio of true positives to predicted positives) is high for both models at 0.99 for both, while the recalls were lower for the 400 Pa predictions at 0.42, and 0.74 at 10,000 Pa. Therefore, the model, despite showing some robustness, should be built for specific pressures in the absence of the Henry coefficient.

**Effects of humidity**. As is the case with other adsorbents[40], the influence of moisture should also be considered in MOFs. Although there are strategies to remove the negative effects of humidity including pre-treatment of feed gas, surface treatment of sorbent, and binder selection, a successful candidate that is inherently not affected by moisture is ideal. As had been highlighted by Kumar et al., the electrostatic interaction under atmospheric conditions, due to a higher partial pressure of $H_2O$, may result in it being preferentially adsorbed over $CO_2$[11]. For the MOFs analyzed in this paper, where the influence of electrostatic interactions clearly plays a significant role (as highlighted in EPoCh descriptors), this will act as an additional challenge when searching for MOF candidates that capture $CO_2$.

Although in this dataset, descriptors regarding the presence of functional groups were not included, it is not unreasonable to suggest that future datasets could include this. Studying MOFs with unusual $CO_2$ affinity at low pressure, Burtch et al. determined that the increase of non-polar functional groups on the benzene dicarboxylate linker of pillared DMOF-1 structure can effectively tune the $CO_2$ Henry coefficient. Particularly, the methyl groups provided the greatest $CO_2$ selectivity over $N_2$, $CH_4$, and CO in relation to other functional groups[41].

Using the threshold of $1.0 \times 10^{-5}$ for the $H_2O$ Henry coefficient for classifying hydrophobic MOFs, suggested by Gulcay et al.[42], 249 MOFs are identified that could be considered hydrophobic (see Fig. 10a). In the reduced dataset, the atom-wise averaged EPoCh descriptors at 40 Pa had a median value of zero and a maximum value below 0.4. This is in contrast to the full dataset where the median value was 0.034 and the maximum was as high as 1.22.

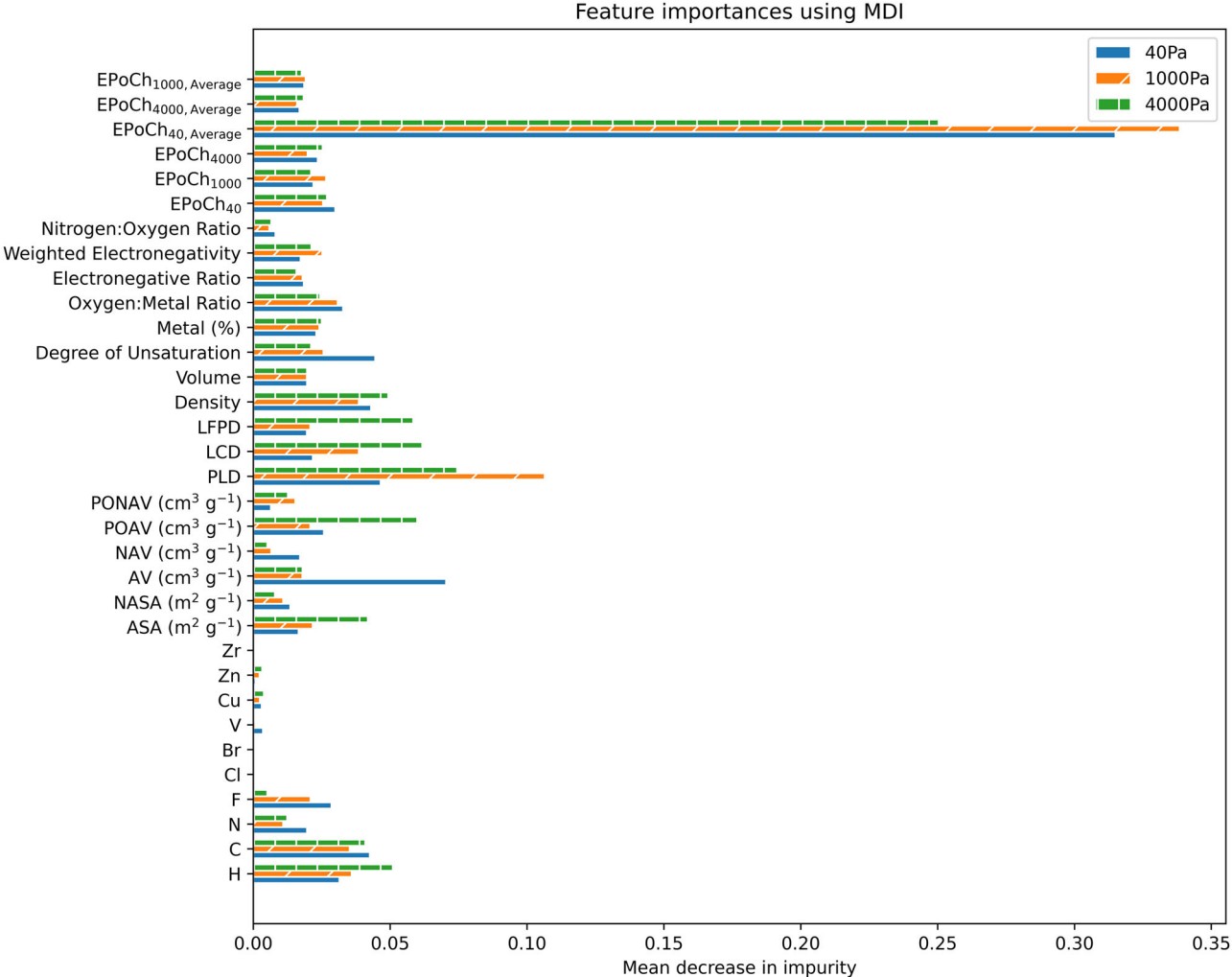

**Fig. 9 Relative importance of descriptors in the model excluding the Henry coefficient of $CO_2$.** Relative importance of descriptors used to build the ML model based on the mean decrease in impurity while traversing nodes of trees in the random forest.

The Henry coefficient of $CO_2$ for this subset of MOFs was similarly lower, obtaining a median of $2.43 \times 10^{-5}$ with a maximum of $9.56 \times 10^{-4}$; while the complete dataset had a median Henry coefficient of $CO_2$ of $2.64 \times 10^{-4}$ with a maximum of $1.175 \times 10^7$. These characteristics correlated with lower uptakes of 0.04, 0.62, and 1.77 mmol g$^{-1}$ in this subset of hydrophobic MOFs at 40 Pa, 1, and 4 kPa, respectively. Using a stricter threshold of $2.6 \times 10^{-7}$, as proposed by Qiao et al.[43], only 71 MOFs met the desired hydrophobicity criterion. The maximum uptakes were obtained at 4 kPa with values reaching 0.411 mmol g$^{-1}$. When compared to the 7.93 mmol g$^{-1}$ adsorption obtained in the complete dataset, the challenges caused by $H_2O$ interactions in limiting MOF candidates for $CO_2$ capture become evident.

Comparing purely the Henry coefficient values of $CO_2$ and $H_2O$, and selecting those that have a greater Henry coefficient for $CO_2$ than $H_2O$ (Fig. 10b), the list of candidate MOFs grows to more than 600. For this set of MOFs the median and 75th percentile uptakes are still lower than those of the remaining dataset; 0.075 and 0.395 mmol g$^{-1}$ compared to 0.229 and 1.211 mmol g$^{-1}$, respectively, for the dataset combining all pressures considered. 206 datapoints in this subset displayed uptakes >1 mmol g$^{-1}$ only 7 of which were at 40 Pa. Four of these seven belonged to the SIFSIX family of MOFs, where SIFSIX-3-Cu displayed the greatest adsorption of 2.49 mmol g$^{-1}$. This uptake value at 40 Pa value corresponds closely with experimental $CO_2$ uptake at 0.1 bar

reported in the literature[44]. Displaying adsorption only 10% lower than SIFSIX-3-Cu was the BUSQIQ MOF from the CoRE MOF dataset. At 1 kPa, the number of MOFs with a $CO_2$ uptake >1 mmol g$^{-1}$ grew to 32 and at 4 kPa this number had reached 167. At 1 and 4 kPa, the $CO_2$ adsorptions had maximums of 4.841 mmol g$^{-1}$ (LOGBEO) and 5.332 mmol g$^{-1}$ (SIHLUQ), respectively. An expanded list of these MOFs can be found in Supplementary Tables 2.1–2.3.

While ML has been shown effective for predicting $CO_2$ uptake, it is evident that the interactions with moisture are another aspect that would eliminate candidate materials based on whether $CO_2$ is being captured through DAC. The ML algorithm has not been directly applied for the purposes of predicting $H_2O$ uptake due to a lack of data pertaining directly to the uptake of $H_2O$. Such a model, if developed, would allow a multi-faceted approach to the screening of MOFs. The combined screening method, using ML for both $H_2O$ and $CO_2$, would then accelerate the screening process even further.

## Conclusions
Looking beyond the conventional descriptors such as those in geometrical and atom-type groups, it was shown that the predictive capabilities could be significantly improved by broadening the scope of descriptors used. In particular, the Henry coefficient

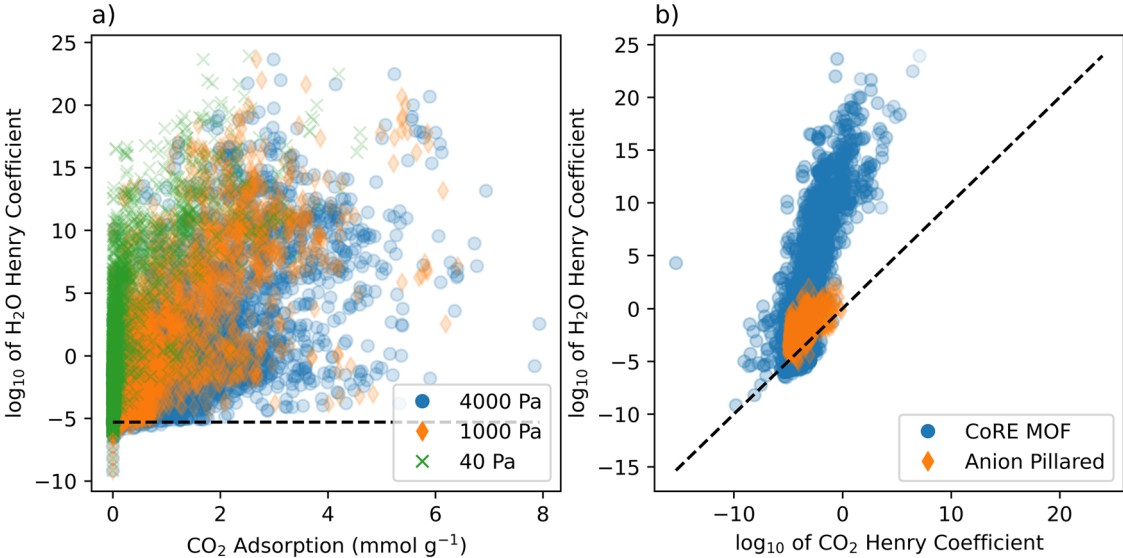

**Fig. 10 Hydrophobicity considerations in the structural dataset. a** Henry coefficient for $H_2O$ plotted against absolute $CO_2$ adsorption where the dashed line is the $1.0 \times 10^{-5}$ threshold where hydrophobic MOFs sit below this line. **b** Henry coefficient for $CO_2$ plotted against the Henry coefficient for $H_2O$, where the dashed line indicates a 1:1 ratio.

was the most influential descriptor in predicting absolute $CO_2$ adsorption; while the EPoCh descriptors could be useful additions for other ML models as they carry information that has not been captured in the benchmark descriptors. The adsorption capacity for $CO_2$ from DAC was shown to be reliably predicted using the features discussed.

Since the equation for gathering EPoCh descriptors for $CO_2$ has been completed, gathering the descriptors for additional MOFs is an expedient process. It is as simple as iterating through each atom's charges in a CIF file and summing the results of the function (Eq. (2)) at the desired pressure. As both the Henry coefficient and EPoCh descriptors require the partial charges on atom sites, the EPoCh descriptors have the benefit of not requiring additional simulations, as opposed to the Henry coefficient which requires each MOF to be run through a Widom insertion simulation.

The hydrophobicity of MOFs remains an important consideration. By comparing the Henry coefficients between $CO_2$ and $H_2O$, 14% of the MOF candidates have a higher affinity for $CO_2$ compared with $H_2O$. Strategies to reduce the effects of humidity are highly encouraged. Alternatively, future ML models could incorporate the effects of humidity to help identify hydrophobic candidates.

In the context of accelerating the discovery of candidate materials, the EPoCh descriptor provides models with significant information while being orders of magnitude faster than gathering the Henry coefficient of the same MOFs. The use of the EPoCh descriptors can therefore accelerate the discovery of new MOFs for DAC and other low partial pressure applications.s

## Data availability
The $CO_2$ uptake of the MOFs used in this paper and their descriptors are available on accessible on GitHub (https://github.com/ibarisorhan/MOF-CO2) and in Supplementary Data 1. The details of the EPoCh descriptor, additional ML performance metrics, relevant MOFs post-screening for hydrophobicity, and gas molecule parameters for RASPA are available in the Supplementary Information.

## Code availability
Jupyter Notebooks, Python scripts, and RASPA simulation input templates used are readily available at: github.com/ibarisorhan/Epoch-Descriptors. github.com/ibarisorhan/MOF-CO2.

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

## Acknowledgements
I.B.O. acknowledges the funding support from the RMIT—CSIRO PhD Scholarship, and all authors acknowledge the National Computing Infrastructure (NCI) and CSIRO Pearcey cluster and Pawsey supercomputing facilities for the computational resources.

## Author contributions
I.B.O. contributed conceptualization, methodology, code development, formal analysis, investigation, data curation, and writing of the original draft. T.C.L., R.B. and A.W.T. contributed conceptualization, software, validation, supervision, and writing/proof-reading of the original draft.

## Competing interests
The authors declare no competing interests.
