## [Peer Review File · Communications Chemistry]

Reviewers' comments:

Reviewer #1 (Remarks to the Author):

Orhan et al. simulate CO₂ adsorption in a set of MOFs, devise and compute cheap descriptors for those MOFs, then train a random forest to predict the CO₂ adsorption in MOFs on the basis of these descriptors. They propose two new descriptors, one based on the potential energy surface inside the MOF, the other based on simulation of fictitious charges. They find the Henry coefficient to be by far the most important feature for the machine learning model, which to me is unsurprising because they are simulating adsorption under low partial pressure of CO₂, pertaining to direct air capture, where Henry's law is likely to hold anyway.

I would categorize this as an exploratory paper that devises two new features of MOFs and empirically studies how different combinations of features produce different predictive performance when used in a supervised machine learning model. I like how the authors quantified the time to compute the descriptors.

I have a few comments for the authors to address before publication.

MAJOR

ONE. Why did you select the anion pillared MOFs? Please 1. Explain and 2. Justify/mention as a caveat that: claims about ML descriptors being predictive or not should be based on a diverse set of structures. This study seems tailored to one particular type of MOFs... I think this is important to discuss since the premise of the paper is that these feature importances hold for other MOFs too. Is the reason for choosing this subset that the charges were available for these? Note, the DDEC MOF data set by Sholl has charges too.

TWO. The stochastic surface sampling descriptor.

TWO A. what is the reasoning for the invention of this descriptor? I found it very odd to look at "gas deflecting off of the surface". What do you mean by this? What is the physical situation here? First, the MOF is an infinite periodic crystal in the simulation. It does not include an actual gas phase in contact with the outer surface of a MOF crystallite. Second, the GCMC pertains to equilibrium properties. This descriptor is motivated by kinetic properties. These two points make it look like the authors have a fundamental mis-understanding about the GCMC simulation. But, the descriptor does contain information about the potential energy surface, so it is okay to include as a descriptor. It would be more reasonable/natural if it were diffusion you were looking at here (which you later mention). At least, justify more carefully why you are investing this descriptor. It just seems odd to me. Why not just bin the potential energy into a histogram, like the Snurr group does? The potential energy is what the GCMC simulation directly relies upon; the GCMC simulation will not account for kinetic hopping like what is happening here in this descriptor. Yes, there are translation moves in the simulation but these are not literal translations in the temporal sense. It is just a way to explore the state space. One could also just do insertions and deletions (no translations) and get the same answer from GCMC (after more samples though, since sampling would be more inefficient).

TWO B. What slide(s) of the MOF did you choose and how, referring to Figure 1? Please explain clearly.

THREE. What is the performance if you apply Henry's law? So, no machine learning model. That should be added as a baseline. The message may be: often simple physics-based approximations can give better performance than a machine learning model.

FOUR. EPoCh descriptors.

FOUR A. You are approximating the function $f(Q, p)$ with equation 2.

Why use a polynomial approximation? This is a machine learning task as well, to learn/approximate $f(Q, p)$ from data. But you choose an arbitrary high order polynomial that can easily overfit. Why not use the random forest for this too? Unclear. That would avoid the zero-ing out you are doing too.

FOUR B. "EPoCh descriptors have the benefit of not requiring a simulation". I thought you used a simulation.

FOUR C. "single atom frameworks were created". Are periodic boundary conditions applied? Then how do you choose the length of the unit cell?

FOUR D. So you get a single number to represent a MOF then? Give intuition for why you are adding them up. I see why but to be clear explain your reasoning. Instead of just "we did this" explain the reasoning behind why/ what your assumption is/ what ideal situation this corresponds to.

FOUR E. Say that "f" is the uptake in the simulation.

FIVE. A huge table of numbers in Table 2? Please make a data visualization. Few have time/patience to stare at your numbers and figure out trends. Poor way to present results. You could turn this into a bar plot for the different metrics to more easily parse/send the message of trends.

ABSTRACT

EPoCh not explained.

INTRODUCTION

Anderson "synthesized". Not really. These are computational models of crystal structures.

"three concentrations of this gas will be considered" replace "this gas" with the specific gas.

METHODOLOGY

"Drop in entropy". Aren't you doing regression? Entropy is the basis for a split for classification, not regression...

Please put in the main text the water and CO₂ model you used and add the appropriate citation. It doesn't make sense to say the model you used for the MOF but not the adsorbate.

"surface was fitted to the simulation results" you gave the resulting function not the parametric function to which the data was fitted...

"No descriptor displayed a significant negative correlation". A negative correlation would be HELPFUL for the machine learning algorithm too. Only "uncorrelated" is not helpful.

12,637 datapoints. Be specific what a “data point” is.

Table 2, numbers are too big for time required. How about using hrs?

Figure 4, 5. X- y-axis labels. CO₂, H₂O need subscripts in plots. This looks low quality. Orange points are buried.

Reviewer #2 (Remarks to the Author):

Orhan, Babarao and Thornton introduced two new descriptor concepts for the prediction of CO₂ adsorption uptake values at low-pressure (40 Pa, 1 kPa and 4 kPa). Although the performance of the model is rather good, it mainly requires the use of the Henry’s constant as a primary descriptor. Given the limited influence of the EPoCh and the S3 descriptors on the overall performance of the final model (see Table 2 and figure 5), I have some doubts on the relevance of introducing them with this uptake prediction task. For me it would make more sense if they were introduced with an ML model that really relies on these descriptors to perform well.

To improve the current work, I suggest clarifying some major points (given below), which are mainly on a comparison of the ML model to a simple Henry’s law and on the relevance of the descriptors in the prediction task. I think the authors should clear up the purpose of the article and the overall consistency between the newly introduced descriptors and the ML task to solve.

Beyond these issues, these new descriptors could potentially be promising to respectively include electrostatic effects and transport effects in an ML model, but this is yet to be proven by further works. Consequently, this work could provide new ideas for future works in the field of adsorption properties prediction involving electrostatic interactions and of diffusion properties prediction.

Major comments:

- The model performs well only when given the Henry's constant (Table 2), we need to make sure that we are not over-complicating a simple problem that could be solved using a simple physical laws. I would recommend comparing the results to the uptake obtained by using the Henry’s law ($N = K_H * P$). Are we out of the Henry's law regime? If we are out of the Henry regime, we need to be more specific on the role of the other descriptors in measuring the gap between the simple Henry law’s uptake and the GCMC calculated uptake.
- Stochastic Surface Sampling models the transition from a unitcell to an adjacent one from the Van der Waals interactions point of view. I am a bit uncomfortable with the use of the terms “exterior” and “interior”, since it could be misinterpreted as if the gas is going out of the material. Please reformulate.
- In the A group descriptors, the choice of the number of atoms per unitcell seems to be a questionable descriptor since doubling the size of the unitcell does not affect the adsorption isotherms but it does affect this descriptor. I would suggest using either a percentage of atoms or a volumetric measure (any intensive variable would be better).
- The scikit learn mean_squared_error is less interpretable than the RMSE in mmol/g. Please put a root on every MSE values given. I would suggest recalculating all the values given in Table 2. using the RMSE (mmol/g) metric.
- If we don’t use Henry’s constants, the models are not very interesting since the MSE is around 0.37,

which corresponds to RMSE of 0.6 mmol/g (in the 1kPa case). These are very high errors for uptake values ranging from 0 to 7 mmol/g (in the 1kPa case). I therefore suggest being more nuanced on the “pivotal role” of EPoCh features in absence of Henry’s constant.

- Since the authors ambition to use a single transferable model to predict any pressures at the low-pressure limit, it would be interesting to predict the uptake at pressures between 40Pa and 1kPa (400 Pa for example) without having seen examples of it in the training set. This would give a better idea on the transferability of the model to different but close values of pressure.

Majorish comments:

- By only reading the Methods, I did not understand how the Effective Point Charge (EPoCh) descriptor was specific to a structure. I would suggest adding some sentences on the fact that Q corresponds to the sum of charges of a given structure and this is how the descriptor is structure-specific. I would also suggest clarifying if these descriptors are absolute values, volumetric or per atom (by reading the Table 1, it suggests all types are present).

- Considering the figure 5, it is very misleading to state that EPoCh descriptors are the second most important group of descriptors. They contribute very little to the overall accuracy of the ML model. Please provide a bit more nuance.

- In the conclusion, the authors concede that S3 descriptors are not “greatly influential” and that they are more suitable for diffusion applications. Please elaborate more on why in the conclusion or/and in the discussions.

- The S3 descriptor could be interesting for diffusion prediction. However, the diffusion transition points are not necessarily placed on the faces of the unitcell (see <https://pubs.acs.org/doi/10.1021/acs.jctc.8b01255>). The choice of the surface seems a bit arbitrary.

- I think there is an error in the Abstract. If I look at Table 2, the D+E descriptors are only hundreds or thousands of times faster than the descriptor F and not “hundreds of thousands of times”. In the conclusion, the authors mention “multiple thousands of times faster”. Please correct it in the abstract.

- I can’t see the general conclusion of the “Interactions with H₂O” section, what is the aim of the data presented? Please clarify this section by stating clearly if the approach works better or worse than the literature.

Minor comments:

- The references are placed inconsistently (spacing, before or after the full point)

- MOFs instead of MOFS

- « the. ML mode. » -> the ML model

- “Not only have increases in CO₂ concentrations have been shown to have impacts on the climate; there are potential negative effects of this increased CO₂ on mammalian physiology” -> “Not only has the increase in CO₂ concentrations been proven to have impacts on the climate, but it also has potential negative effects on mammalian physiology”

- “remains to be capture” -> “remains to be captured”

- “that that” -> “than that”

- “H₂O” don't forget the subscript on 2

- “the cutoff distance for interactions between set to” words are missing

- “to modelled” -> “to model”

- The data is easily accessible via a github repository and the Raspa simulations are perfectly

reproducible given the details provided. However, details on the EPoCh and S3 descriptors are yet to be uploaded.

Reviewer #1

1. Reviewer Comment: Why did you select the anion pillared MOFs? Please 1. Explain and 2.

Justify/mention as a caveat that: claims about ML descriptors being predictive or not should be based on a diverse set of structures. This study seems tailored to one particular type of MOFs... I think this is important to discuss since the premise of the paper is that these feature importances hold for other MOFs too.

Is the reason for choosing this subset that the charges were available for these? Note, the DDEC MOF data set by Sholl has charges too.

Author Response:

Two datasets were chosen and combined for this study. Firstly, the CoRE MOF 2019 dataset was chosen with a diverse range of features and readily available charges (Kancharlapalli, Gopalan et al. 2021). Secondly, the Anion-Pillared dataset was chosen due to exceptional CO₂ capacity at low partial pressures. Both datasets were used together in building the machine learning model. While only the Anion-Pillared MOFs were used for determining the time necessary to collect the descriptors. We have added the following to clarify this in the main manuscript (Section Dataset Curation on page 3):

In this study, MOFs from the CoRE MOF dataset (3,378 structures) and the Anion-pillared MOF dataset (936 structures) were used, where partial charges on the atom sites had been calculated based on DFT using the DDEC method [31, 32]. While both datasets were used in the ML model, only the Anion-pillared MOFs were used to estimate the necessary time for gathering descriptors.

2. Reviewer Comment: The stochastic surface sampling descriptor.

(A) What is the reasoning for the invention of this descriptor? I found it very odd to look at “gas deflecting off of the surface”. What do you mean by this? What is the physical situation here? First, the MOF is an infinite periodic crystal in the simulation. It does not include an actual gas phase in contact with the outer surface of a MOF crystallite. Second, the GCMC pertains to equilibrium properties. This descriptor is motivated by kinetic properties. These two points make it look like the authors have a fundamental mis-understanding about the GCMC simulation. But, the descriptor does contain information about the potential energy surface, so it is okay to include as a descriptor. It would be more reasonable/natural if it were diffusion you were looking at here (which you later mention). At least, justify more carefully why you are investing this descriptor. It just seems odd to me. Why not just bin the potential energy into a histogram, like the Snurr group does? The potential energy is what the GCMC simulation directly relies upon; the GCMC simulation will not account for kinetic hopping like what is happening here in this descriptor. Yes, there are translation moves in the simulation but these are not literal translations in the temporal sense. It is just a way to explore the state space. One could also just do insertions and deletions (no translations) and get the same answer from GCMC (after more samples though, since sampling would be more inefficient).

(B) What slide(s) of the MOF did you choose and how, referring to Figure 1? Please explain clearly.

Author Response:

(A) The reasoning behind the two new descriptors, EPoCh and S3, was to isolate the effects of the electrostatic interactions and the VDW forces, respectively. The results show that the EPoCh descriptor is good at predicting uptake, while the S3 descriptor showed poor correlation with uptake. Upon reflection, we have decided to remove the S3 descriptor from the manuscript. However, we believe this S3 descriptor will be useful for predicting diffusion/kinetic properties and will be a topic of future work.

(B) S3 descriptor has been removed from the manuscript and will appear in future work.

3. Reviewer Comment: What is the performance if you apply Henry's law? So, no machine learning model. That should be added as a baseline. The message may be: often simple physics-based approximations can give better performance than a machine learning model.

Author Response:

Yes, we thank the reviewer for the suggestion of using Henry's Law to evaluate the performance of the ML model. We show that Henry's Law is a poor predictor for uptake. A coefficient of determination -9.807×10^{15} was observed for the entire dataset. This was due to a large number of MOF's having high Henry coefficient's (K_H) resulting in a significant over prediction, up to 3 orders of magnitude higher than the GCMC result. If we remove structures with a Henry coefficient > 0.05 , then the law was reasonable for low pressures (40Pa and 1000Pa) with $R^2 \approx 0.9$. However, the trends began to diminish at higher pressures (4000Pa). Furthermore, structures with low Henry coefficient's < 0.05 are not good candidates due to their low uptakes at low partial pressures.

A new section on "Henry's Law" has now been added to the main manuscript (see Page 8).

4. Reviewer Comment: EPoCh descriptors:

(A) You are approximating the function $f(Q, p)$ with equation 2.

Why use a polynomial approximation? This is a machine learning task as well, to learn/approximate $f(Q, p)$ from data. But you choose an arbitrary high order polynomial that can easily overfit. Why not use the random forest for this too? Unclear. That would avoid the zero-ing out you are doing too.

(B) "EPoCh descriptors have the benefit of not requiring a simulation". I thought you used a simulation.

(C) "single atom frameworks were created". Are periodic boundary conditions applied? Then how do you choose the length of the unit cell?

(D) So you get a single number to represent a MOF then? Give intuition for why you are adding them up. I see why but to be clear explain your reasoning. Instead of just "we did this" explain the reasoning behind why/ what your assumption is/ what ideal situation this corresponds to.

(E) Say that "f" is the uptake in the simulation.

Author Response: We appreciate these questions, we have addressed and included clarifications where appropriate in the main manuscript.

(A) A polynomial was selected to 1) ensure the exact replicability of the values gathered, 2) obviate the necessity for additional libraries (such as scikit-learn), and 3) reduce time and complexity in predicting MOF candidates. Considering the smoothness of the simulated uptake around an isolated point charge, we found that a polynomial surface was capable of adequately capturing this effect. Overfitting is not a concern here since the values evaluated are desired to either directly match one of the simulated values or to follow the trends as closely as possible. From this equation, the effects of partial charge and partial pressure are captured together in one descriptor. Therefore, a polynomial function that takes in the arguments of pressure and charge was seen as appropriate.

We have added a new Figure 1 and discussion on Page 7 to further explain this descriptor.

(B) Simulations were run to evaluate the effects of point charges in isolation. These simulations created a dataset from which the polynomial equation was derived and there is no need for additional simulations for gathering the EPoCh descriptor for CO₂ after this. The sentence identified by the reviewer was referring to any additional simulations. The effects of the partial charges within the MOF frameworks' atoms, when evaluated as isolated point charges, can therefore be evaluated directly through the polynomial equation instead of running additional GCMC simulations for each partial charge.

To address the confusion that may arise from the wording, the phrase “not requiring a simulation” has been changed to “not requiring additional simulations” (see 2nd paragraph in the “Conclusions” section).

(C) In this implementation, the size was set to the nearest multiple of 10 greater than the VDW interaction distance in simulations: a cube with 20 Angstrom lengths in each dimension. The periodicity settings are then applied by default. The “atom” was a fictional entity that was massless and carried no VDW forces. This detail is now included in Section Effective Point Charge (EpoCh) Descriptors on Page 6.

(D) Yes, the aim is to have a single number to represent the effect of electrostatic charges in a MOF. However, a collection or family of numbers was shown to be more effective. As such we evaluated each partial charge in the framework, through the polynomial, and then averaged the results according to both atomic density and volumetric density. This was performed for each MOF. To avoid confusion in the manuscript, the text was adjusted accordingly, see Section Effective Point Charge (EpoCh) Descriptors on Page 7:

The partial charges on the atoms are unique to each framework and by evaluating each charge of a framework through Equation 2, their isolated effects are estimated. The results are averaged both volumetrically and atom-wise, to determine a suite of EPoCh descriptors for each MOF structure.

(E) We have added the following text to clarify “f” (see Page 6):

The surface was fitted to the simulation results, where f is the uptake, using the following equation:

$$f(Q, p) = \alpha_1 Q + \alpha_2 Q^2 + \alpha_3 Q^3 + \alpha_4 Q^4 + \alpha_5 Q^5 + \alpha_6 Q^6 + \alpha_7 Q^7 + \alpha_8 p + \alpha_9 p^2 + \alpha_{10} p^3 + \alpha_{11}$$

5. Reviewer Comment: A huge table of numbers in Table 2? Please make a data visualization. Few have time/patience to stare at your numbers and figure out trends. Poor way to present results. You could turn this into a bar plot for the different metrics to more easily parse/send the message of trends.

Author Response: We agree with the reviewer’s suggestion and have reduced the amount of data, moved the table to the supporting information and have summarised the data in new figures. See Machine Learning Results section and Figure 5 on Page 11:

Figure 5. Coefficient of determination R^2 for the ML models. A+B+C is the benchmark model using conventional descriptors. The addition of the EPoCh descriptors (D) and the Henry coefficient energy descriptors (E) shows an improvement in the model.

Reviewer #2

Major Comments:

1. Reviewer Comment: The model performs well only when given the Henry's constant (Table 2), we need to make sure that we are not over-complicating a simple problem that could be solved using a simple physical laws. I would recommend comparing the results to the uptake obtained by using the Henry's law ($N = K_H * P$). Are we out of the Henry's law regime? If we are out of the Henry regime, we need to be more specific on the role of the other descriptors in measuring the gap between the simple Henry law's uptake and the GCMC calculated uptake.

Author Response: In our response to a similar question from Reviewer #1, we show that Henry's Law is a poor predictor of uptake. While MOFs with a low K_H abide to the law, the increases in pressure show a diminishing adherence to the trend. In particular, the top performing MOFs are no longer in the linear-uptake region when plotted against pressure. To address the applicability of Henry's Law a new section was added to the manuscript. It was observed that MOFs with a higher Henry coefficient were not able to accurately be modelled through this law and it was these MOFs that have such a high proclivity to capturing CO₂ that they have already surpassed the "linear region" of the isotherm where Henry's law holds true. These are the MOFs that we are most interested. While a handful of MOFs with high uptake are still present even when these are screened out, screening based on Henry Coefficient alone therefore is not accurate enough to be used as a direct classification method for uptake. This can be observed in our new Figure 4 (see Page 10):

Figure 4. GCMC uptake versus Henry's Law uptake for (a) MOFs with Henry's coefficients below or equal to 0.001, and (b) MOFs with Henry's coefficients between 0.001 and 1. The dashed black lines indicate perfect agreement between GCMC and Henry's Law.

2. Reviewer Comment: Stochastic Surface Sampling models the transition from a unitcell to an adjacent one from the Van der Waals interactions point of view. I am a bit uncomfortable with the use of the terms "exterior" and "interior", since it could be misinterpreted as if the gas is going out of the material. Please reformulate.

Author Response: Based other comments from Reviewer #1, we have removed the S3 descriptor from the manuscript. It was not a good predictor of uptake and is more suited for future work on diffusion properties.

3. Reviewer Comment: In the A group descriptors, the choice of the number of atoms per unitcell seems to be a questionable descriptor since doubling the size of the unitcell does not affect the

adsorption isotherms but it does affect this descriptor. I would suggest using either a percentage of atoms or a volumetric measure (any intensive variable would be better).

Author Response: Yes these features have been converted to a volumetric basis and the corresponding models have been updated. The results in the manuscript now reflect these changes with no changes in the final outcome and Table 1 is now updated.

4. Reviewer Comment: The scikit learn mean_squared_error is less interpretable than the RMSE in mmol/g. Please put a root on every MSE values given. I would suggest recalculating all the values given in Table 2. using the RMSE (mmol/g) metric.

Yes we agree with the reviewer. The mean square errors have been changed to root mean square errors accordingly. See Section 6 in Supplementary Information for the RMSE data.

5. Reviewer Comment: If we don't use Henry's constants, the models are not very interesting since the MSE is around 0.37, which corresponds to RMSE of 0.6 mmol/g (in the 1kPa case). These are very high errors for uptake values ranging from 0 to 7 mmol/g (in the 1kPa case). I therefore suggest being more nuanced on the "pivotal role" of EPoCh features in absence of Henry's constant.

Author Response: We agree with the reviewer and are grateful for their recommendation. While the regression values provide lower R^2 values and higher RMSE values in the models without K_H , the regions into which the predictions fall when doing pseudo-classification are comparable for both models. The value had been set to 1 mmol/g for pseudo-classification; however, to verify this claim, direct classification models were trained at each pressure for various thresholds. The performances by these models have been shared as a supporting document to further substantiate this claim. The key role that EPoCh descriptors play comes from the speed through which they are obtained. Being able to classify MOFs at faster speed than models including K_H (due to the time requirements of collecting the descriptors), while obtaining comparable values will be the key role of these descriptors.

See new additions in the manuscript on this topic on Page 14.

6. Reviewer Comment: Since the authors ambition to use a single transferable model to predict any pressures at the low-pressure limit, it would be interesting to predict the uptake at pressures between 40Pa and 1kPa (400 Pa for example) without having seen examples of it in the training set. This would give a better idea on the transferability of the model to different but close values of pressure.

Author Response: Yes, we agree with the recommendation by the reviewer. To address this, the MOFs within the dataset were predicted at 400Pa to be within 40Pa and 1kPa as per the recommendation, and at 10 kPa to evaluate how the model performs beyond the pressures on which it was trained. The combined model was trained by adding pressure as an additional descriptor. The results have been added to the text in the manuscript (see Page 15):

In this study, separate ML models were created for each pressure considered. While this is beneficial for evaluating MOFs where there is already data available, the capacity of MOFs at pressures unseen by the model are not predictable through separate models. By adding "Pressure" as a descriptor and combining all the datapoints to a single ML model, predictions were made on MOFs at pressures where there were previously no data. Two pressures were selected for testing predictions by interpolation (400 Pa) and extrapolation (10,000 Pa). 400 Pa was selected so that the model could make predictions between pressures on which it has been trained (40 Pa and 1,000 Pa). To see how it performs when considering pressures beyond the maximum (4,000 Pa), 10,000 Pa was selected. At 400 Pa the model yielded an R^2 of 0.54 while at 10,000 Pa the R^2 was 0.72

when the Henry coefficient was excluded from the dataset, see results in Supplementary Information Figure S6.1. Applying the same pseudo-classification threshold of 1 mmol/g, the precision (ratio of true positives to predicted positives) is high for both models at 0.99 for both, while the recalls were lower for the 400 Pa predictions at 0.42, and 0.74 at 10 000 Pa. Therefore, the model, despite showing some robustness, should be built for specific pressures in the absence of the Henry coefficient.

Figure S6.1. Predicted uptakes vs simulated uptakes for pressures unseen to the ML model.

Secondary Comments:

1. Reviewer Comment: By only reading the Methods, I did not understand how the Effective Point Charge (EPoCh) descriptor was specific to a structure. I would suggest adding some sentences on the fact that Q corresponds to the sum of charges of a given structure and this is how the descriptor is structure-specific. I would also suggest clarifying if these descriptors are absolute values, volumetric or per atom (by reading the Table 1, it suggests all types are present).

Author Response: Yes, the EPoCh Section has been changed to clarify this matter with text highlights indicating where changes were made. The following passage, relevant to this issue, was added:

The partial charges on the atoms are unique to each framework and by evaluating each charge of a framework through Equation 2, their isolated effects are estimated. The results are averaged both volumetrically and atom-wise, to determine a suite of EPoCh descriptors for each MOF structure..

2. Reviewer Comment: Considering the figure 5, it is very misleading to state that EPoCh descriptors

are the second most important group of descriptors. They contribute very little to the overall accuracy of the ML model. Please provide a bit more nuance.

Author Response: Thank you for highlighting the possibility of misinterpretation of the statement on which the feedback was given. The wide range of the K_H values causes the model to be prone to over-emphasising its importance; in the model where K_H is present, there is a range of features competing for second highest importance, with all falling significantly below the importance of K_H .

The averaged EPoCh descriptor has the second greatest importance listed. The model built excluding K_H shows clearly that the most influential descriptor is from the EPoCh family, along with others that are ranked high in importance.

Figure 8. Relative importance of descriptors in the model excluding the Henry coefficient of CO₂

In terms of the RMSE and R^2 values obtained in the models without K_H , we agree that the accuracy still suffers from the absence of K_H . When considering the purpose of these models, which is to act as a screening process to determine which MOFs would be worth further analysis, the models without K_H (that rank EPoCh as the most important) display similar pseudo-classification performances.

See additional discussion on Page 14.

3. Reviewer Comment: In the conclusion, the authors concede that S3 descriptors are not “greatly influential” and that they are more suitable for diffusion applications. Please elaborate more on why in the conclusion or/and in the discussions.

Author Response: S3 descriptor has been removed and will be considered in future work.

4. Reviewer Comment: The S3 descriptor could be interesting for diffusion prediction. However, the

diffusion transition points are not necessarily placed on the faces of the unitcell (see <https://pubs.acs.org/doi/10.1021/acs.jctc.8b01255>). The choice of the surface seems a bit arbitrary.

Author Response: S3 descriptor has been removed and will be considered in future work.

5. Reviewer Comment: I think there is an error in the Abstract. If I look at Table 2, the D+E descriptors are only hundreds or thousands of times faster than the descriptor F and not “hundreds of thousands of times”. In the conclusion, the authors mention “multiple thousands of times faster”. Please correct it in the abstract.

Author Response: Yes, there is some confusion here that has been resolved. It is true that the “D+E” combination is only multiple thousands of times faster than gathering K_H , however, the EPoCh descriptor is more than 1M times faster (based on averages for the Anion-Pillared MOFs). A new Figure 7 highlighting regarding the time requirement for each feature was added:

Figure 7. Estimated times necessary to gather descriptors of a 10,000 MOF dataset (based on timings gathered from the Anion-Pillared MOFs).

6. Reviewer Comment: I can't see the general conclusion of the “Interactions with H₂O” section, what is the aim of the data presented? Please clarify this section by stating clearly if the approach works better or worse than the literature.

Author Response: We have added text in the conclusion to make this section more coherent. While the method acts primarily as an expedient screening method, the conclusions are in regards to the sheer number of MOFs that could be eliminated by the method of screening for hydrophobicity. The interactions with H₂O remain an important consideration for MOFs and would benefit from having its own ML model. As such, the following text was added:

While ML has been shown effective for predicting CO₂ uptake, it is evident that the interactions with moisture are another aspect that would eliminate candidate materials based on whether CO₂ is being captured through DAC. The ML algorithm has not been directly applied for the purposes of predicting H₂O uptake due to a lack of data pertaining directly to the uptake of H₂O; such a model, if developed, would allow a multi-faceted approach to the screening of MOFs. The combined

screening method, using ML for both H₂O and CO₂, would then accelerate the screening process even further.

REVIEWERS' COMMENTS:

Reviewer #1 (Remarks to the Author):

The authors have improved their manuscript by removing the poorly-motivated descriptor from their analysis and comparing to the baseline Henry coefficient model.

I recommend publication, but only after the authors provide all data to reproduce their manuscript.

Currently, the Github repository is not self-contained. That is, the files needed to run the notebook are not there. Please add the data so that the study can be reproduced. This is not acceptable.

https://github.com/ibarisorhan/EPoCh-Descriptors/blob/main/EPoCh_Github.ipynb

If it is unclear why having open data and a reproducible project is necessary, please see:

<https://pubs.acs.org/doi/10.1021/acs.chemmater.7b00799>

and

<https://www.nature.com/articles/s41557-021-00716-z>

Reviewer #2 (Remarks to the Author):

The authors have thoroughly answered the different questions/concerns I had. I therefore recommend publication.

I just have an additional question that does not require any change in the manuscript. I was just wondering if the authors have any explanations on how the EPoCh descriptor helps to predict the GCMC uptake. What is the information that it gives in addition to Henry constants ? (Just for scientific curiosity)

Reviewer #1

Reviewer Comment:

The authors have improved their manuscript by removing the poorly-motivated descriptor from their analysis and comparing to the baseline Henry coefficient model.

I recommend publication, but only after the authors provide all data to reproduce their manuscript. Currently, the Github repository is not self-contained. That is, the files needed to run the notebook are not there. Please add the data so that the study can be reproduced. This is not acceptable.

https://github.com/ibarisorhan/EPoCh-Descriptors/blob/main/EPoCh_Github.ipynb

If it is unclear why having open data and a reproducible project is necessary, please see:

<https://pubs.acs.org/doi/10.1021/acs.chemmater.7b00799>

and

<https://www.nature.com/articles/s41557-021-00716-z>

Author Response:

The complete datasets and algorithms are now available through the Github repository. Additionally, the MOF dataset is provided as supplementary information.

Reviewer #2

Reviewer Comment:

The authors have thoroughly answered the different questions/concerns I had. I therefore recommend publication.

I just have an additional question that does not require any change in the manuscript. I was just wondering if the authors have any explanations on how the EPoCh descriptor helps to predict the GCMC uptake. What is the information that it gives in addition to Henry constants ? (Just for scientific curiosity)

Author Response:

The EPoCh descriptors provide additional information about the adsorption behaviour specifically around charged atoms, as opposed to the Henry constant that considers both van der Waals and partial charges. Furthermore, the EPoCh descriptors are calculated at different pressures while the Henry constant is calculated at infinite dilution. The following paragraph was added to revised manuscript on Page 9:

In comparison with the Henry coefficient, the EPoCh descriptors provide additional information about the adsorption behaviour specifically around charged atoms. The Henry coefficient gives an overall picture of the interactions for CO₂ uptake in the isotherm's linear region at infinite dilution. For example, all interactions are captured in that number, and by multiplying it by the pressure, we can obtain the uptake. However, some high-uptake MOFs quickly (almost immediately) fall outside the linear region. The EPoCh descriptor, on the other hand, is indifferent to which region of the MOF's isotherm it falls as it is calculated at different pressures. Unlike K_H , it does not capture all interactions; it captures only the electrostatic interactions at the pressures considered, ignoring the VDW interactions. Overall, the ML model is improved by the charged atoms' electrostatic interactions being modelled more precisely.